# Non-Canonical Splicing and Its Implications in Brain Physiology and Cancer

**DOI:** 10.3390/ijms23052811

**Published:** 2022-03-04

**Authors:** Consuelo Pitolli, Alberto Marini, Claudio Sette, Vittoria Pagliarini

**Affiliations:** 1Department of Neuroscience, Section of Human Anatomy, Catholic University of the Sacred Heart, 00168 Rome, Italy; consuelo.pitolli@unicatt.it (C.P.); claudio.sette@unicatt.it (C.S.); 2GSTEP-Organoids Research Core Facility, IRCCS Fondazione Policlinico Universitario Agostino Gemelli, 00168 Rome, Italy; alberto.marini26@gmail.com

**Keywords:** back-splicing, microexons, chimeric RNAs, recursive splicing, RNA-binding proteins, brain tumours

## Abstract

The advance of experimental and computational techniques has allowed us to highlight the existence of numerous different mechanisms of RNA maturation, which have been so far unknown. Besides canonical splicing, consisting of the removal of introns from pre-mRNA molecules, non-canonical splicing events may occur to further increase the regulatory and coding potential of the human genome. Among these, splicing of microexons, recursive splicing and biogenesis of circular and chimeric RNAs through back-splicing and trans-splicing processes, respectively, all contribute to expanding the repertoire of RNA transcripts with newly acquired regulatory functions. Interestingly, these non-canonical splicing events seem to occur more frequently in the central nervous system, affecting neuronal development and differentiation programs with important implications on brain physiology. Coherently, dysregulation of non-canonical RNA processing events is associated with brain disorders, including brain tumours. Herein, we summarize the current knowledge on molecular and regulatory mechanisms underlying canonical and non-canonical splicing events with particular emphasis on cis-acting elements and trans-acting factors that all together orchestrate splicing catalysis reactions and decisions. Lastly, we review the impact of non-canonical splicing on brain physiology and pathology and how unconventional splicing mechanisms may be targeted or exploited for novel therapeutic strategies in cancer.

## 1. Introduction

Almost all human genes are transcribed and spliced in order to generate mature transcripts before being translated into proteins or to carry out specific regulatory functions as non-coding RNAs [1]. The splicing process is mediated by a large macromolecular complex, named the spliceosome, which coordinates and catalyses splicing reactions [1]. The spliceosome machinery is able to recognize the generally short exonic sequences that are interspersed between long introns. Since the sequences that mark the exon–intron boundaries display degenerate nature with relatively little sequence constraints at the splice sites, numerous cis-acting elements and trans-acting factors assist the spliceosome in the definition of the coding exons to be included in the mature transcript [2]. Competition between splice sites, differential recruitment of splicing factors and/or RNA-binding proteins (RBPs), DNA and RNA modifications, chromatin and RNA secondary structures, kinetics of transcriptional elongation and intron removal, all contribute to regulation of splicing, greatly expanding the number of transcript variants arising from a single gene locus through a process termed alternative splicing [3]. Interestingly, these regulatory mechanisms greatly impact splicing efficiency, thus contributing, at least in part, to unconventional splicing events that can occur alternatively or concomitantly with canonical splicing events (see below). Importantly, although canonical and non-canonical splicing utilize the same spliceosome machinery and many of the associated splicing factors and RBPs, the two RNA maturation processes are not always governed by the same rules [4]. The central nervous system (CNS) represents the tissue in which unconventional splicing events are most common, possibly reflecting, or more likely, contributing to its functional complexity [5,6,7,8]. Thus, it is not surprising that deregulation of non-canonical splicing strongly correlates with dysfunctional programs of neuronal differentiation and brain development [5,9], and ultimately, with brain disorders [5]. Herein, we discuss the molecular mechanisms underlying canonical and non-canonical splicing events, with particular emphasis on the impact of unconventional splicing events on brain physiology and pathology.

## 2. Mechanisms of Canonical Splicing: Splicing and Alternative Splicing

In eukaryotes, the vast majority of precursor messenger RNAs (pre-mRNAs) are composed of coding sequences (exons) interrupted by non-coding sequences (introns) [10]. Therefore, intronic sequences need to be removed and surrounding exons joined, to generate a mature and translatable messenger RNA (mRNA). Interestingly, the majority of long non-coding RNAs (lncRNAs) also share this discontinuous structure and undergo removal of introns to generate the mature transcript. This process of RNA maturation, which affects more than 90% of human genes, is termed splicing [11].

### 2.1. Molecular Mechanisms Underlying Splicing Catalysis

The splicing process requires conserved regulatory sequences located within introns: a GU dinucleotide at the 5′ end of the intron (5′ splice site or donor site); a branch point (BP) adenosine and an AG dinucleotide at the 3′ end of the intron (3′ splice site or acceptor site); a poly-pyrimidine tract upstream of the 3′ splice site [12]. These regulatory sequences act as docking sites for five small-nuclear ribonucleoprotein particles (snRNPs U1, U2, U4, U5, U6) that represent the core spliceosome that catalyses the process [1,13]. Each of these splicing particles is composed of a single small nuclear RNA (snRNA U1, U2, U4, U5, U6), a set of seven similar proteins (Sm-proteins) and additional proteins that are specific for each particle [1,13].

Spliceosome assembly starts with the recognition of the 5′ splice site by the U1 snRNP. This interaction is partially mediated by the sequence-specific recognition of the 5′ splice site by the U1 snRNA [1,13]. Other non-snRNP proteins, including the branch point binding protein (BBP) SF1, which binds the BP site, and the heterodimeric complex U2 snRNP auxiliary factor (U2AF35/65), which recognizes the poly-pyrimidine tract and the 3′ splice site, take part in the early steps of spliceosome formation [1,13]. At this stage, bridging interactions occur between U1 snRNP and SF1/U2AF, thus bringing into close proximity the 5′ and 3′ ends of the intron and defining it for excision. The formation of this complex, termed complex E, commits pre-mRNA to the splicing catalysis. The progression of the complex E toward the next complex A is triggered by the base pairing-mediated binding of U2 snRNPs to the BP site. U2AF plays an important role at this stage by stabilizing U2 snRNPs-BP interaction [1,13]. In the last step of spliceosome formation (complex B), U4/U6 and U5 snRNPs are incorporated into the spliceosome as a preformed tri-snRNP, triggering several rearrangements in the RNA–RNA interactions, that culminate with the formation of the complex C of the spliceosome and with the splicing reaction [1,13].

Splicing occurs through two consecutive transesterification reactions [1,13]. In the first step, the 2′-OH group of the BP adenosine attacks and cleaves the phosphodiester bond at the 5′ splice site leading to the formation of a looped structure (lariat loop) as result of a bond between the first intronic nucleotide and the BP adenosine. In the second step, the newly formed 3′-OH at the end of the exon attacks and cleaves the phosphodiester bond at the 3′ splice site. The result of this reaction is the joining of the 5′ splice site of the upstream exon with the 3′ splice site of the downstream exon and the excision of the intron included between them. At the end of the splicing reaction, the spliceosome disassembles and is recycled for new splicing processes [1].

Splicing efficiency can be affected by the size of the exons and the introns. In higher eukaryotes, most genes are characterized by short exons separated by longer introns [14]. Thus, spliceosome assembly preferentially occurs across an exon rather than across an intron (exon definition model) since pairing splice sites across a small exon appears easier that across a very large intron [15]. Conversely, in lower eukaryotes, in which genes are typically characterized by small intron and large exons, the spliceosome prevalently assembles across introns (intron definition model) [15]. The efficiency of splice site recognition can be also affected by exon size through steric hindrance. Indeed, the distance between splice sites must be sufficient to impair steric interference between spliceosome factors involved in their recognition. Supporting this hypothesis, skipping can be induced if the exon size is reduced below 50 nucleotides [16].

### 2.2. Alternative Splicing

Since sequences at the exon/intron boundaries are highly degenerate in higher eukaryotes [12], additional regulatory sequences in the pre-mRNA are required to improve identification of the correct splice sites by the spliceosome. These RNA elements include exonic and intronic splicing enhancers (ESEs and ISEs) or silencers (ESSs and ISSs) that act by either promoting or inhibiting the recognition of splice sites by spliceosome, respectively [2]. ESEs and ISEs act as binding sites for splicing factors, such as serine/arginine-rich proteins (SR proteins), that promote the use of a nearby splice site by directly recruiting the splicing machinery or counteracting the action of nearby silencer elements. In contrast, ESS and ISS elements are bound by splicing repressors, such as the heterogeneous nuclear ribonucleoproteins (hnRNPs), that inhibit the positioning of spliceosome at the splice sites. The interplay between such antagonistic splicing factors can determine whether an exon (cassette exon) or part of it (alternative 5′ and 3′ splice sites) is included or not in the mature mRNA through a process named alternative splicing [2]. Indeed, while many exons are constitutively spliced, nearly all human genes also contain variant exons that can be alternatively spliced, allowing each human gene to encode multiple splice variants and proteins. The flexible regulation of the alternative splicing process allows cells to rapidly adapt to changes in the surrounding environment or to progressively acquire specific functions during differentiation through expression of specific splice variants, thus greatly enhancing the coding potential of eukaryotic genomes [2].

### 2.3. Regulation of Alternative Splicing

Transcription and splicing are connected in time and space [17] and most transcripts are spliced co-transcriptionally [18,19], and in some cases, coordinated by the transcription machinery. Two models have been proposed to explain how transcription can impact splicing [17,20]. The first model, known as the “recruiting model”, based on the ability of the RNA polymerase II (RNAPII) to recruit splicing factors to the nascent transcripts, thus favouring splice site recognition and splicing. Indeed, the carboxy-terminal domain (CTD) of the large RNAPII subunit contains heptapeptide (YSPTSPS) repeats [21]. With the exception of the two prolines, all other residues of this heptapeptide repeat sequence can be dynamically phosphorylated during the transcription cycle by several kinases, including the transcription-associated cyclin-dependent kinases (CDKs) [22]. The phosphorylation state of the CTD can modulate recruitment of specific splicing factors and spliceosome assembly. For instance, phosphorylation of the CTD on the serine 2 residue is required to recruit U2AF65, and in turn, U2 snRNP [23]. According to the second model, termed “kinetic coupling model”, changes in the elongation rate of RNAPII can influence splicing by modulating splice sites’ selection. In particular, slow transcriptional rate results in more time for weak splice site recognition compared to a faster rate, promoting exon inclusion [24,25].

In addition to the kinetics of RNAPII, epigenetic chromatin modifications can also affect alternative splicing acting as scaffold to recruit splicing factors or as roadblocks to slow down the RNAPII elongation rate [26,27,28].

## 3. Mechanisms of Non-Canonical Splicing

### 3.1. Back-Splicing and Alternative Back-Splicing

Unlike canonical pre-mRNA splicing, in which the 5′ splice site of an upstream exon is joined with the 3′ splice site of a downstream exon, back-splicing is an event in which the 5′ splice site of a downstream exon is joined with the 3′ splice site of an upstream exon, thus yielding a circular transcript (circRNA) [29]. Most of the annotated circRNAs are exclusively composed by exons (exonic circRNAs—EcircRNAs), but these molecules may also derive from introns (intronic circRNAs—IcircRNAs) or contain both intronic and exonic sequences (exonic–intronic circRNAs—EIciRNAs). Interestingly, since circRNAs do not expose free ends, they are highly resistant to the action of exonucleases, have longer half-life compared to linear RNAs and tend to accumulate in tissue during aging [29]. This feature allows circRNAs to be efficiently detected in body fluids, such as blood or urine, and support their potential as biomarkers [30].

#### 3.1.1. Molecular Mechanisms Underlying circRNAs Biogenesis

Two models have been proposed to explain circRNA biogenesis [29]. In the first model, known as lariat loop intermediate, the process starts with a pre-mRNA transcript that undergoes a canonical alternative splicing event. The result of this process is the generation of a properly spliced linear mRNA transcript concomitantly to a lariat intermediate containing the exonic and/or intronic sequences, which have been skipped as result of the splicing catalysis. In turn, the lariat intermediate can be further back-spliced to generate a circular molecule [29]. In the second model, termed direct back-splicing, the back-splicing process comes first, generating a circRNA concomitantly with an exon–intron–exon linear intermediate that can be further processed [29]. Although it is still unclear which between these models is more effective in circRNAs formation, it is known that both the recognition of the canonical splice sites and the activity of the spliceosome machinery are required for an efficient back-splicing event [31].

#### 3.1.2. Cis-Acting Elements and Trans-Acting Factors Involved in circRNAs Biogenesis

Additional cis-acting regulatory elements and trans-acting factors positively regulate circRNAs biogenesis. Among the cis-acting regulatory elements, inverted Alu elements and smaller complementary sequences have been clearly shown to play a functional role in circRNAs biogenesis [32,33] (Figure 1A). Pairing of inverted Alu repeats and/or complementary sequences in introns that flank the exons involved in the back-splicing reaction contributes to circRNAs biogenesis by bringing into close proximity a downstream 5′ splice site and an upstream 3′ splice site [33,34] (Figure 1A). The proximity of back-splicing sites may be also guaranteed by self-dimerization of specific RBPs, such as Muscleblind (MBL) [35] and Quaking (QKI) [36]. These RBPs were shown to bind specific sequence elements located in the introns flanking the back-spliced exon/s, and their dimerization favoured the formation of a loop between these introns that, in turn, promotes pre-mRNA circularization (Figure 1A). This is also true for the neuron-specific RBP named NOVA alternative splicing regulator 2 (NOVA2) [37]. NOVA2 binding sites have been identified in introns flanking circularizing exons, and their presence was essential to permit back-splicing events to occur [37]. Even if the functional role of NOVA2 self-dimerization in circRNAs biogenesis has not been fully investigated [37], it is likely that NOVA2-dependent back-splicing events also rely on its ability to form homodimers.

Interestingly, dimerization of RBPs and pairing of inverted Alu repeats can also cooperate in circRNAs biogenesis. For instance, Sam68, a member of the signal transduction and activation of RNA (STAR) family of RBPs such as QKI [38], has been reported to promote the circularization of the survival motor neuron 1/2 (SMN1/2) pre-mRNA [39]. Sam68 binding sites are located in very close proximity or within Alu elements that are in inverted orientation in introns flanking the SMN exons involved in circularization. In vitro mutagenesis studies showed that deletion of the Sam68 binding sites or Alu elements in only one of the two introns involved in back-splicing events partially impaired SMN circRNAs biogenesis. However, deletion of both regulatory elements completely abrogated the formation of the SMN circRNA, suggesting the functional role of Sam68 in promoting and/or strengthening the formation of the Alu-mediated double-stranded RNA that is necessary for the circularization of the transcript [39].

Among the negative regulators of circRNA biogenesis, the DNA and RNA helicase DExH-box helicase 9 (DHX9) and the RNA-editing enzyme adenosine deaminase RNA specific 1 (ADAR1) represent excellent examples in this sense [40,41]. DHX9 represses circRNA biogenesis by unwinding the double-stranded RNA loops formed by inverted repeat sequences, thus limiting the contact between the splice sites involved in the back-splicing event [41]. On the other hand, ADAR1 is an editing enzyme which converts adenosine to inosine editing, thus reducing the complementarity between inverted repeat sequences and inhibiting circRNAs biogenesis [40].

Similar to canonical splicing, back-splicing is also subjected to regulation. The competition between inverted complementary sequences across introns that bracket the circRNA-forming exons can give rise to multiple alternative circular transcripts from the same gene locus through a mechanism termed alternative back-splicing [34].

#### 3.1.3. The Complex Crosstalk between Canonical Splicing and Back-Splicing

Although back-splicing is less frequent (<1%) than canonical splicing [42], some circRNAs are ubiquitously expressed, and in some cases, generated more efficiently than their linear counterpart [8,43].

In particular, back-splicing becomes more prevalent when canonical splicing is inhibited/impaired [44]. For instance, depletion or pharmacological inhibition of the core spliceosome factor splicing factor 3b subunit 1 (SF3B1) increased the expression levels of circRNAs, and concomitantly decreased the expression of their linear counterparts [44]. A similar effect on circular/linear RNA ratio is observed upon depletion of the pre-mRNA 3′-end processing factor cleavage and polyadenylation specific factor 3 (CPSF73). In this case, the limiting amounts of CPSF73 lead to inhibition of transcription termination due to formation of readthrough transcripts, which undergo back-splicing events and generate numerous circRNAs [44]. Another important piece of experimental evidence of the coupling between splicing efficiency and transcript circularization was provided by the observation that back-splicing propensity is positively correlated with the elongation rate of RNA pol II [35,42], which is known to change the outcome of many splicing events [17].

Back-splicing and canonical pre-mRNA splicing can also regulate each other by competing for the same splicing sites. Indeed, when an exon prone to circularization is flanked by exons carrying strong 5′ and 3′ splice sites, pre-mRNA is mainly processed by canonical splicing. However, mutation of these sites to weaken their strength caused an increase in circularization of the transcript, suggesting that linear splicing of flanking exons can compete with circRNA biogenesis [35]. Based on this well-documented competition between linear splicing and back-splicing, even the biogenesis of the circMBNL RNA, arising from a back-splicing event of the second exon of the muscleblind (MBNL) transcript, is favoured at the expense of its linear counterpart [35]. Interestingly, this competition between linear and circular splicing of the MBNL transcript is finely orchestrated by MBNL protein itself, which specifically binds to the introns flanking exon 2 and favours its circularization [35]. Thus, in addition to identifying a novel regulator of circRNAs biogenesis, this study highlighted a novel mechanism of regulation of gene expression, based on competition between linear and circular splicing, through which the intracellular amount of MBNL protein controls its own expression by increasing the levels of circMBL at the expense of the linear MBNL transcript [35].

Recently, the splicing factor proline and glutamine-rich (SFPQ) protein has also been shown to play an important role in orchestrating competition between canonical- and back-splicing [45]. SFPQ binding is specifically enriched in the introns flanking the so-called DALI circRNAs, which are characterized by distal inverted Alu elements and long flanking introns [45]. Depletion of SFPQ expression strongly affected DALI circRNAs biogenesis as result of usage of cryptic splice acceptor sites, intron retention and premature polyadenylation and transcription termination [45]. Interestingly, depending on whether the cryptic splice acceptor site is located in the upstream or downstream introns that flank the circularizing exon, the biogenesis of DALI circRNAs is favoured or inhibited, respectively. In fact, when the cryptic acceptor site, utilized upon SFPQ depletion, is located in the upstream intron, the acceptor splice site of circularizing exon is unspliced and available for back-splicing events as result of the competition between cryptic and canonical splice sites [45]. Thus, since SFPQ dysfunction is correlated with diverse neurological diseases, such as amyotrophic lateral sclerosis (ALS) [46,47] and frontotemporal lobar degeneration (FTLD) [48], it is possible to speculate an essential role for this RBP in guaranteeing proper splicing of long introns and thus in maintaining the correct homeostasis of circRNAs in brain physiology and pathology.

#### 3.1.4. Cellular Functions of circRNAs

Many circRNAs are conserved between species and expressed in a tissue- and developmental stage-specific manner, suggesting their potential regulatory functions [8,43,49]. Although a growing number of circRNAs with functional significance is emerging [50], providing an exhaustive list of them is outside the scope of this review. Herein, we will illustrate a few representative examples of circRNAs for which specific biological functions have been demonstrated in detail.

One of the molecular mechanisms through which circRNAs regulate gene expression consists of sequestering (sponging) microRNAs (miRNAs) preventing their binding and their regulatory activity on the 3’ UTR regions of their target genes [51]. However, circRNAs and miRNAs show very different quantities within the cell, with circRNAs usually expressed in lower copy numbers compared to miRNAs. Indeed, circRNAs, which act as efficient natural sponges for miRNAs, usually contain a high number of miRNA binding sites, thus guaranteeing the proper stoichiometry between the two interacting molecules [52]. The brain-specific antisense to the cerebellar degeneration-related protein 1 transcript (CDR1as, also called circular RNA sponge for miR-7, ciRS-7) and the testis-specific sex-determining region Y (Sry) circRNAs represent excellent examples in this sense, harbouring >70 and 16 binding sites for miR-7 and miR-138, respectively [53,54]. Alternatively, albeit with a reduced number of binding sites for single miRNAs, some circRNAs can perform their function of sponging by binding several miRNAs at the same time. This is the case for circHIPK3, an abundant circRNA derived from exon 2 of the homeodomain-interacting protein kinase 3 (HIPK3) gene [55]. It regulates the proliferation of different human cell lines by sponging nine different miRNAs with 18 potential binding sites [55]. Thus, by inhibiting the activity of several miRNAs together, some circRNAs, such as circHIPK3, have great potential to regulate a vast repertoire of cellular processes that together may contribute to the tumorigenic phenotype of cancer cells.

In addition to interacting with miRNAs [51], circRNAs can also act as sponges of RBPs. This is the case for two different circRNAs, circPCNX and circPABPN1, which affect stability and translation of the cyclin-dependent kinase inhibitor 1A (CDKN1A) and autophagy-related gene 16L1 (ATG16L1) transcripts, respectively, thus impacting on two pathways (i.e., proliferation and autophagy) that are essential to guaranteeing cellular homeostasis [56,57]. Interestingly, these circRNAs showed opposite effects on the translation of their downstream targets through modulation of the activity of the RBP with which they interacted [56,57]. In particular, circPCNX sequestered the AU-rich element RNA-binding protein 1 (AUF1) [57], which is known to mediate the degradation of most of the transcripts to which it binds [58], thus increasing the stability and the translation of CDKN1A [57]. On the other hand, circPABPN1 sponged the human antigen R (HuR) protein [56], which is known to positively regulate translation of its target genes [59], thus decreasing the translation of ATG16L1 without affecting its stability [56]. Another circRNA able to affect transcript translation is circBACH [60]. This circRNA, like circPABPN1, also binds to HuR, but in this case, acts as a carrier of the RBP rather than sponging it, thus promoting HuR translocation to the cytoplasm [60]. CircBACH1-dependent relocation of HuR resulted in the suppression of internal ribosome entry site (IRES)-dependent translation of cyclin-dependent kinase inhibitor 1B (CDKN1B), and consequently, in inhibition of cellular proliferation [60,61]. Through a similar molecular mechanism but with a very different result, circZNF609 (zinc finger protein 609) binds to HuR (alias ELAV-like RNA-binding protein 1 ELAVL1) and selectively loads it onto the cytoskeleton-associated protein 5 (CKAP5) mRNA with which circZNF609 directly interacts [62]. CircZNF609-dependent recruitment of HuR on the CKAP transcript positively regulates its stability and translation, strongly impacts tumour growth, and sensitises cancer cells to microtubule-targeting chemotherapeutic drugs [62]. CircRNAs have also been shown to regulate the stability of proteins by directly interacting with them. In this context, it was shown that circNDUFB2, generated from exons 2 and 3 of the NADH:ubiquinone oxidoreductase subunit B2 (NDUFB2) gene, impaired non-small-cell lung cancer progression by promoting the degradation of the oncogenic insulin-like growth factor 2 mRNA-binding proteins (IGF2BPs) [63]. Mechanistically, circNDUFB2 facilitated tripartite motif-containing protein 25 (TRIM25)-mediated ubiquitination and degradation of IGF2BPs by acting as a scaffold molecule between these two proteins and enhancing their interaction [63].

Interestingly, circRNAs can also regulate the expression of their own linear transcripts by recruiting RNA pol II on their parental gene promoter [64]. In this context, a special class of intron-retained circRNAs (EIcircRNAs), localized into the nucleus, were shown to interact with U1 snRNP by base-pairing with U1 snRNA [64]. EIcircRNA-U1 snRNP complexes, in turn, recruited RNA Pol II on parental gene promoters and enhanced their expression [64]. Furthermore, experimental evidence suggests a functional role of some circRNAs in recruiting epigenetic modifiers on specific gene loci in order to regulate their expression, further highlighting the multiple layers at which circRNAs can modulate biological functions [65].

Although circRNAs are usually classified as non-coding RNA, some of them contain an internal ribosome entry site (IRES) that can be recognized by the eukaryotic initiation factor 4G2 (eIF4G2) to drive translation of peptides [66,67,68]. However, not all translated circRNAs contain IRES sequences; in this case, translation starts through a mechanism that requires circRNAs post-transcriptional modifications. In particular, adenosine residues within short RNA elements [RRACH (R=G or A; H=A, C or U)], which are highly enriched in circRNAs, can be methylated in the form of N6-methyladenosines (m6A) [69]. Such modified motifs display IRES-like activities and can allow for translation of circRNAs. This process involves the recruitment of the m6A reader protein YTH N6-methyladenosine RNA-binding protein 3 (YTHDF3) which, in turn, recruits the translation machinery [69]. An excellent example in this sense is represented by circZNF609, a circular transcript that originates from the circularization of the second exon of its host gene [67]. Interestingly, circZNF609 owns an open reading frame (ORF) and is translated in a cap-independent manner [67]. In particular, methyltransferase 3 (METTL3)-dependent m6A modification of circZNF609 mediated the recruitment of the m6A reader YTHDF3 together with the translational initiation factor eIF4G2 on circZNF609, thus favouring its translation [70]. However, as endogenous circRNAs-derived peptides are difficult to detect and most have only been observed following in vitro overexpression experiments, the coding potential of most circRNAs and the positive regulatory role of m6A methylation on their translation need to be further investigated [67,71,72,73].

### 3.2. Splicing of Microexons

Microexons are defined as very small exons (3–30 nt) that can be included in the mRNA sequence as result of non-canonical alternative splicing [74,75]. Compared to exons of regular size, the efficient splicing of a microexon presents some critical aspects. Indeed, based on the exon definition model (see above) microexons do not have the optimal size to avoid steric interference between spliceosome components bound to the 5′ and 3′ splice sites. Moreover, their short length limits the number of ESE that can be contained in their sequence. However, the analysis of microexons and of their flanking regions revealed features that allow them to compensate for inefficient exon definition [76]. In particular, compared to exons of regular size, microexons are characterized by shorter flanking introns (suggesting a more efficient intron definition) and stronger 5′ and 3′ splice sites that facilitate their recognition [76]. Moreover, introns flanking microexons are particularly enriched in motifs for specific RBPs, such as RNA-binding Fox-1 homolog 1 (RBFOX1), PTBP1 and serine/arginine repetitive matrix 4 (SRRM4) which regulate their splicing (Figure 1B) [5,76]. Interestingly, recent experimental evidence has highlighted the involvement of RBPs known to regulate polyadenylation process—a maturation process of the pre-mRNA consisting of cleavage and subsequent addition of a poly(A) tail to an RNA transcript—such as the cytoplasmic polyadenylation element-binding protein 4 (CPEB4) [77], CPEB2, CPEB3 and factor interacting with PAPOLA and CPSF1 (FIP1L1) [78]. However, for most of these RBPs the molecular mechanism/s underlying microexons splicing has/have not been fully elucidated, except for SRRM4. SRRM4 binds specialized upstream intronic enhancer elements together with the serine/arginine rich splicing factor 11 (SRSF11) and RNA binding protein with serine rich domain 1 (RNPS1) thus, to posi-tively regulate the inclusion of microexons into mature mRNAs [79,80]. Notably, the C-terminal of SRRM4 contains an enhancer of microexons (eMIC) domain, which mediates its interaction with the branchpoint-binding protein SF1 and U2AF, thus promoting recruitment of U2 snRNP and permitting the definition of the microexon [81].

Functionally, microexon inclusion can have two possible outcomes. If the microexon is in-frame, its inclusion in the mature mRNA can add few amino acids to the encoded protein, thus providing novel functional properties [76]. On the other hand, if the microexon is not in frame, it mostly leads to the introduction of a premature stop codon, and consequently, to the degradation of the resulting transcript through the nonsense-mediated decay (NMD) pathway [76]. Interestingly, recent evidence has shown a possible functional role for the endogenous exon junction complex (EJC), a protein complex that is deposited on the spliced RNA at ∼20–24 nt upstream of each exon–exon junction, in regulating the splicing of microexons (Figure 1B) [82,83]. Indeed, the SR-related protein RNPS1, which was shown to regulate splicing of some neuronal microexons (see below), associates with the EJC to mitigate production of microexons from cryptic splice sites (Figure 1B) [80,82].

### 3.3. Trans-Splicing

Trans-splicing is a non-canonical RNA processing event that generates chimeric transcripts by joining together exons of two separated pre-mRNA molecules derived either from the same gene (intragenic trans-splicing) or from different genes (intergenic trans-splicing (Figure 1C) [84]. Recently, a new class of trans-spliced chimeric transcripts (cross-strand chimeric RNA) has been identified, which are generated by the fusion between RNAs transcribed by the two opposite DNA strands [85].

The resulting chimeric RNA may either encode for novel proteins with mostly uncharacterized functions or it may act as non-coding RNA with mostly regulatory roles in gene expression. Thus, trans-splicing, like alternative splicing and back-splicing, might represent an evolutionary strategy aimed to increase transcriptome and proteome diversity.

#### 3.3.1. Molecular Mechanisms Underlying Chimeric RNAs Biogenesis

The discovery of this process changed the assumption of chromosomal translocation as the only source of chimeric transcripts [86]. Indeed, chimeric RNA biogenesis can also occur as a co-transcriptional or post-transcriptional event due to non-canonical processing of immature RNA transcripts.

Since trans-splicing requires the recognition of the same cis-acting regulatory elements that govern the canonical splicing reactions, it also depends on the same repertoire of splicing factors and RBPs to occur. In particular, according to the spliceosome-mediated trans-splicing model, the two unprocessed transcripts, which are involved in trans-splicing event, are recruited to the same spliceosome machinery and joined at canonical “GU-AG” sites [87]. However, trans-splicing events involving atypical splice donor/splice acceptor dinucleotides have also been identified [88]. An example is provided by the rat leukocyte common antigen-related (LAR) transcript, whose alternative 3′ UTR is generated by a trans-splicing event involving a non-canonical GC-TC site [88].

An additional model, known as transcriptional slippage model, has also been proposed to explain trans-splicing-mediated chimeric RNA formation. This model suggests that the elongation complex of RNAPII involved in the transcription of one gene can move to transcribe a second gene, thus generating a chimeric pre-mRNA transcript [89]. Interestingly, the presence of small (<10 bp) homology sequences (SHSs) in the parental genes of chimeric transcripts was shown to favour the base-pairing between the first transcript and the DNA of the second one, thus permitting slippage of the transcriptional machinery [89]. These SHSs are essential for chimeric RNA generation, as demonstrated by the impairment of this process following their disruption by mutation [89].

In both proposed models, the 3D proximity between the two genes involved in the trans-splicing process is a necessary prerequisite for the event to take place [90]. In this context, an interesting study observed that the parental genomic regions of many chimeric transcripts are enriched in binding motifs for specific DNA-binding factors, such as the CCCTC-binding factor (CTCF) [91,92], which are able to bring parental genes into close proximity within discrete sites into the nucleus where coordinated transcription occurs (named transcriptional factories) [93], thus promoting the biogenesis of chimeric RNA transcripts. In light of these observations, the enrichment of specific regulatory elements, such as SHSs and/or CTCF binding sites, might be predictive of a high propensity of certain genes to undergo trans-splicing.

#### 3.3.2. Examples of Chimeric RNAs in Humans

An ancestral type of trans-splicing, known as leader sequence (SL) trans-splicing, is a very common process in lower species, such as trypanosomes and nematodes. A short leader sequence is spliced to the 5′ end of almost all transcripts and is required to guarantee mRNA stability, transport and translation [94]. Although there is no evidence of the existence of SL trans-splicing in higher eukaryotes, other types of intra- and inter-genic trans-splicing events have been reported in vertebrates [95,96]. Although trans-splicing remains a rare event, these chimeric transcripts have been involved in many physiological and pathological conditions, including human cancers [97,98]. The trans-spliced chimeric RNA juxtaposed with another zinc finger protein 1 (JAZF1)—SUZ12 polycomb repressive complex 2 subunit (SUZ12 alias JJAZ1)—represents an excellent example in this sense [99]. The JAZF1-SUZ12 chimeric RNA is expressed in both normal endometrial stromal cells [100] and in endometrial stromal sarcomas [99,101,102] and it encodes a fusion protein with anti-apoptotic properties [102]. In normal endometrial tissue, the expression of the JAZF1-SUZ12 chimeric transcript is modulated during the menstrual cycle in response to changes in hormone levels [100], with lower expression levels during the menstrual phase and higher levels in the early proliferative and late secretory phases [100]. The higher expression of the JAZF1-SUZ12 transcript in later stages of secretory phase could play a functional role in guaranteeing the viability of stem endometrial cells and in preventing their apoptosis during blood vessel constriction and bleeding that occur in the following menstrual phase [102]. Interestingly, while the JAZF1-SUZ12 chimeric RNA derives from a chromosomal translocation in the tumour context, no evidence of DNA rearrangement was reported in healthy tissue, suggesting that the transcript is generated by a post-transcriptional trans-splicing event [100].

Another example is represented by paired box 3 (PAX3)–forkhead box O1 (FOXO1), a fusion RNA transiently expressed during differentiation of pluripotent muscle cells into skeletal muscle [103,104]. Interestingly, in alveolar-type rhabdomyosarcoma (ARMS), the chimeric PAX3-FOXO1 RNA directs a profound epigenetic chromatin remodelling in cooperation with the master transcriptional regulators of muscle differentiation (MYOD, MYOG and MYCN). These molecules impact the activity of super enhancers and impair muscle differentiation by freezing the cells in a progenitor state [105]. As previously described for JAZF1-SUZ12 chimeric RNA, the PAX3-FOXO1 transcript is generated by a chromosomal translocation exclusively in the tumour context, like ARMS [106], while it is probably generated by a post-transcriptional trans-splicing event in healthy cells where the corresponding chromosomal rearrangement at the DNA level was not observed [103,104]. Interestingly, these observations give rise to the hypothesis known as “the cart before the horse hypothesis”, which proposes that the biogenesis of trans-spliced chimeric RNAs could precede and promote DNA rearrangements underlying cancer-related fusion genes [107]. Such RNA-induced genome rearrangements are a common phenomenon in ciliates [108], but some evidence suggests that chimeric RNAs may facilitate the onset of gene fusions also in mammals [109]. In particular, fusion RNAs may act as template for the repair machinery at DNA double-strand breaks regions or, alternatively, they may intercalate for sequence complementarity in the DNA duplexes of two distant chromosomes, bringing them into proximity and promoting strand breakage and genome rearrangement [110].

### 3.4. Recursive Splicing

Recursive splicing is a non-canonical splicing mechanism used to excise long introns (>24 kb) from pre-mRNA. Unlike canonical splicing in which introns are removed as a single unit, during recursive splicing long introns are removed as smaller segments through two consecutive steps (Figure 1D) [6,7].

First discovered in Drosophila Melanogaster, recursive splicing requires the presence of a motif, the recursive splice (RS) site, containing a 3′ splice site dinucleotide (AG) followed by a 5′ splice site dinucleotide (GT) located within the long intron [111,112]. During the first splicing step, the donor splice site of the upstream exon is joined to the 3′ splice site of the RS site allowing for the removal of the first part of the intron. During the second step, the GT sequence of the RS site acts as donor site and it is spliced with the 3′ splice site of the downstream exon, thus removing the second part of the intron (Figure 1D) [111].

In vertebrates (including humans), long introns may contain a cryptic exon or microexon—depending on its size—termed RS exon, which overlaps with the RS site and contains a partial 5′ splice site motif at its beginning [6]. The presence of the RS exon adds a further layer of regulation because during the second recursive splicing reaction it creates a competition between two alternative 5′ splice sites: the recursive 5′ splice site at the exon-RS exon junction (RS-5ss), reconstituted when the RS exon is spliced to its preceding exon, and the 5′ splice site of RS exon. The competition between these two alternative 5′ splice sites will determine if the RS exon will be included or skipped in the mature mRNA [6,82]. Since the inclusion of RS exons usually introduces an in-frame premature stop codon leading to degradation of the transcript by the NMD pathway, it could represent a quality-control mechanism to remove faulty RNA molecules [6]. However, when it maintains the frame, the inclusion of RS exons could also provide an evolutionary strategy for expanding the coding and regulatory potential of the genome [111].

Despite considerable research efforts, little is known about the molecular mechanisms and about the RBPs involved in the regulation of recursive splicing. Interestingly, recent experimental evidence showed that the EJC plays an important role in regulating this process [80,82,83,113]. Since most functional studies of the EJC have focused on its cytosolic role in premature termination codon recognition during NMD [114], more recently it has been shown to prevent the recognition of cryptic splice sites and re-splicing of mature transcripts into the nucleus [82,83]. Interestingly, EJC deposition also occurs upstream of the exon-RS exon junction generated by the first step of recursive splicing, thus impairing the recognition of RS-5ss and spliceosome assembly. Thus, suppressing the use of splice sites that are generated by newly formed exon junctions capable of recursive splicing, the EJC complex has been identified as a negative regulator of this process and its action strongly contribute to inclusion of annotated RS exons in mature mRNAs [82,83,113]. In support of this evidence, depletion of core components of the EJC, such as the eukaryotic translation initiation factor 4A3 (eIF4A3), RNA-binding motif protein 8A (RBM8A) and MAGO homolog, exon junction complex subunit (MAGOH), or EJC-associated proteins, such as Pinin (PNN) and RNPS1, resulted in widespread splicing deregulation and usage of cryptic and reconstituted exonic 5′ splice sites generated by new exon junctions [82,83]. Interestingly, in the absence of the EJC, these reconstituted exonic 5′ splice sites are re-spliced to the acceptor site of a downstream intron, resulting in exon skipping [82,83]. Moreover, depletion of eIF4A3 expression increased the binding of pre-mRNA processing factor 8 (PRPF8), a core component of the spliceosome, to RS-5ss, further confirming the role of the EJC in inhibiting the recognition of cryptic or reconstituted exonic 5′ splice sites and of the recursive splicing [82]. Coherently, lower levels of EIF4A3 and MAGOH expression in brain [82] might strongly contribute to splicing of neuronal microexons and justify the prevalence of recursive splicing in this tissue [6,75].

In conclusion, by regulating the use of cryptic splice sites, recursive splicing prevents spurious formation of cryptic microexons and contributes to properly shaping the mammalian transcriptome.

## 4. Impact of Non-Canonical Splicing in Brain Physiology

### 4.1. CircRNAs in Brain Physiology

CircRNAs are highly expressed in brain, and especially in the cerebellum, with about 20% of protein-coding genes producing circRNAs [8,49]. Their expression shows developmentally specific patterns, suggesting a potential role for this class of molecules in the regulation of neuronal functions [8,115]. CDRas1 represents an excellent example in this sense, being highly abundant and having important functional implications in the brain [116]. CDRas1 represents the first circular transcript for which a functional role as an miRNA sponge was demonstrated nature [30,53]. In fact, it harbours >70 binding sites for miR-7 and was shown to strongly suppress miR-7 activity, and consequently, to positively regulate the expression of miR-7 target transcripts, thus ensuring proper brain development (Figure 2A) [9,53,54]. Interestingly, morpholino-mediated knockdown of miR-7 in zebrafish induced defects of brain development that were phenocopied by overexpressing CDRas1 into embryos [54]. Moreover, the physiological role of CDRas1 in the brain was further investigated by the development of the first animal model in which the expression of a circular transcript was deleted (Figure 2A) [9]. Although Cdr1as knockout mice are viable and fertile, they suffer from defects in sensorimotor gating, possibly due to altered excitatory synaptic transmission (Figure 2A) [9]. Interestingly, these defects are also observed in some neuropsychiatric disorders and neurodegenerative diseases [117], highlighting the possibility that dysregulation of CDRas1 or other neuronal-enriched circRNAs could contribute to pathological brain contexts.

Another recent example of circRNA with a functional role in the brain is represented by circZNF827. This circRNA acts as a negative regulator of neuronal differentiation by repressing the expression of key neuronal genes, such as nerve growth factor receptor (NGFR) [118]. Mechanistically, circZNF827 was proposed to repress the expression of brain-related genes by recruiting on their promoters a transcriptional repressive complex composed by its own host-encoded protein, ZNF827, and the heterogeneous nuclear ribonucleoprotein K (hnRNP K) and hnRNP L. [118].

Due to their high stability and high resistance to enzymatic degradation, circRNAs tend to accumulate during aging [8,119,120]. This phenomenon is particularly evident in the CNS, a tissue characterized by very low or totally absent proliferation [8,42,119]. A circRNA whose expression was reported to increase during ageing in the brain is circGRIA1, which is encoded by the AMPA receptor GluR1 locus. circGRIA1 expression shows an age-related and male-specific increase in the prefrontal cortex and hippocampus of rhesus macaques [121]. Interestingly, circGRIA1 negatively regulates the expression of its host glutamate ionotropic receptor AMPA-type subunit 1 (GRIA1) gene via its association with the promoter region of the parental genomic locus [121]. CircGRIA1-dependent decreases in glutamate receptor levels strongly affected synaptic plasticity and synaptogenesis [121]. Coherently with the functional role of circGRIA1 in the synaptic function, many other brain-expressed circRNAs are localized in synaptosomes and they are mostly generated from genes encoding for synaptic proteins [8,115]. Moreover, the expression level of many of these synaptic circRNAs is deregulated in response to alterations in neuronal activity [115], suggest that circRNAs could effectively contribute to the biology of brain aging and pave the ground for exciting scenarios in which circRNAs might be also involved in aging-related diseases. Indeed, given their impact on synaptic plasticity and neuronal function, deregulation of circRNAs’ expression in a specific time window during neuronal differentiation and synaptogenesis might underlie the pathogenesis of brain diseases, including neurodegenerative pathologies and cancer.

### 4.2. Splicing of Microexons in Brain Physiology

Inclusion of microexons in mature mRNAs is a particularly frequent event in the CNS, with more than 60% of the regulated microexons preferentially included in neural transcripts [5,76]. Neuronal microexons are frequently controlled by a complex consisting of SRRM4, also known as neural-specific sr-related protein of 100 kda (nSR100), SRSF11 and RNPS1 (Figure 1B), whereas RBFOX1 regulates an overlapping, although distinct and smaller, subset of microexons [5,76,80]. Recent evidence suggests a role for the spliceosomal component pre-mRNA processing factor 40 (PRP-40) and RBP Quaking (QKI) in the regulation of microexons in the CNS [122,123], further expanding the repertoire of factors involved in splicing control of this peculiar class of exons.

Mice haploinsufficient for SRRM4/nSR100 showed splicing dysregulation of specific microexons and displayed phenotypes associated with autism spectrum disorder (ASD), including altered social behaviour, increased sensitivity to environmental stimuli and altered synaptic spine density and transmission [124,125]. Coherently, patients affected by ASD are characterized by very low expression levels of SRRM4/nSR100 and splicing of these microexons is strongly inhibited [5]. Since neuronal microexons encode for residues that are prevalently located on protein surfaces, mostly regulating protein–protein interactions [5], their splicing repression in ASD patients likely contributes to the pathogenesis of the disease by altering protein functions [5]. In support of this hypothesis, an elegant study has recently elucidated the functional role of a neuronal microexon of the eukaryotic translation initiation factor 4 gamma (eIF4G) in the control of higher-order cognitive functioning [126]. When included in the eIF4G mRNA, the microexon favoured the interaction of eIF4G protein with cytoplasmic mRNP granule components, including the fragile X-linked translation regulator (FMRP), thus promoting ribosome stalling on translationally repressed synaptic protein transcripts and preventing new rounds of translation initiation. Conversely, when the microexon is skipped, this repression is relieved and translation of numerous proteins that control synaptic transmission and neuronal activity is upregulated, contributing to an activated neuronal state and strengthening synaptic connectivity [126]. Consistently with this key functional impact, mice in which the expression of eIF4G microexon is genetically deleted showed altered social behaviour, memory and learning deficits. Moreover, analysis of RNA sequencing data revealed that splicing of the eIF4G microexon is disrupted in a subset of autistic patients [5,126], further linking this splicing event to brain physiology.

Although the functional significance of microexons in brain development has not been fully characterized, their expression profiles are dynamically modulated during neuronal differentiation, further supporting their functional role during brain development [5,78]. For instance, the inclusion of a 6 nt microexon in mature amyloid-beta precursor protein binding, family b member 1 (APBB1) mRNA is modulated during differentiation of mouse embryonic stem cells (mESCs) into cortical glutamatergic neurons and strongly affects the properties of the cognate protein [5]. APBB1 protein comprises two phosphotyrosine-binding domains named PTB1 and PTB2, which mediate the interaction with the histone acetyltransferase Kat5/Tip60 and the amyloid-beta precursor protein (APP), respectively [127]. Interestingly, the inclusion of the microexon in the APBB1 mRNA adds two charged residues in the PTB1 domain of the corresponding protein and significantly enhances its interaction with the histone acetyltransferase Kat5/Tip60, and to a lesser extent, with APP [5]. Since loss of Kat5 activity is associated with developmental defects that impact learning and memory [128], a strong interaction between APBB1 and Kat5, mediated by the splicing of a microexon, is most likely functional to ensure proper brain development.

Several other studies have recently highlighted the physiological role of microexons in regulating the functional interaction and the activity of neural proteins involved in neuron-related signalling pathways. Clear examples in this sense are represented by the regulation of the transcripts encoded by the intersectin 1 (ITSN1) [129] and protrudin genes [130]. Splicing of a 15 nt microexon in the neuron-specific ITSN1 transcripts leads to inclusion of five charged amino acid residues within its SH3A domain in the neuronal membrane ITSN1 protein and promotes its interaction with dynamin protein [129] and its activity as adaptor protein and regulator of synaptic vesicles transport [131]. Likewise, inclusion of a 21 nt microexon in the protrudin mRNA strongly enhances the binding affinity of the encoded protein for the vesicle-associated, membrane protein-associated protein (VAP-A), thus promoting the extension of neurites [130].

Interestingly, although inclusion of the majority of neural microexons (80–90%) maintains the mRNA ORF, splicing of others yields an in-frame stop codon and lead to transcript degradation via NMD [5]. This is the case for the pro-apoptotic BCL2 antagonist/killer 1 (BAK1) transcript (Figure 2B). Inclusion of a 20 nt microexon in the mature BAK1 mRNA leads to an in-frame premature stop codon causing NMD-dependent degradation of the transcript and downregulation of BAK1 protein (Figure 2B) [132]. Interestingly, this splicing event is specifically regulated during brain development, and it is required to attenuate apoptosis and to guarantee survival of post-mitotic neurons (Figure 2B) [132]. In immature neurons, the splicing of the BAK1 microexon is strongly repressed by the binding of the PTBP1 splicing factor, whose expression is strongly downregulated during brain development, thus contributing to repression of BAK1 protein expression (Figure 2B) [132].

Other studies have shown the functional role of neuronal microexons in chromatin regulation and transcription [133], axon growth and synapse formation [78,124,134,135], neuronal differentiation [136], microglia homeostasis [122] and animal behaviour [77,133,137]. In conclusion, splicing of neuronal microexons represents an evolutionarily conserved mechanism of gene expression regulation which contributes to the functional complexity of the CNS. In line with its essential role in brain physiology, splicing of microexons is finely regulated and its dysregulation is associated with neurodegenerative diseases and brain disorders [75].

### 4.3. Trans-Splicing in Brain Physiology

Chimeric RNAs have been considered for a long time to be the result of cancer-related chromosomal rearrangements [138]. However, several studies have demonstrated that many chimeric RNAs are expressed in normal tissues, including the brain, likely as result of trans-splicing events [139,140]. Although the functional role of RNA chimeras in the brain has not been fully investigated, some chimeric RNAs regulate cellular processes with important functional implications in brain development, such as neuronal differentiation and pluripotency maintenance [141]. An interesting study identified four intragenic trans-splicing transcripts (originating from casein kinase 1 gamma 3 (CSNK1G3), Rho GTPase activating protein 5 (ARHGAP5), FAT atypical cadherin 1 (FAT1) and rhabdomyosarcoma 2 associated transcript (RMST) loci) that are highly expressed in human pluripotent stem cells and differentially expressed in human embryonic stem cells (hESCs) differentiation, although for none of these, with the exception of tsRMST, has their functional role been further investigated [141]. However, of special interest is the expression profile of the chimeric tsRMST, the first reported trans-spliced large intergenic long non-coding RNA (lincRNA) [141]. Interestingly, the expression of tsRMST is strongly downregulated during hESC differentiation and it is not detected in adult normal human tissues, including the brain, raising the possibility that this trans-spliced transcript plays a role in maintenance of hESC self-renewal and/or early lineage differentiation (Figure 2C) [141]. Indeed, depletion of the tsRMST chimeric transcript strongly impaired the maintenance of hESC pluripotency by suppressing the expression of differentiation-related genes, such as GATA binding protein 4 (GATA4), GATA6, and paired box 6 (PAX6; Figure 2C). Mechanistically, tsRMST contributed to these effects by enhancing the recruitment of the transcription factors NANOG and SUZ12 on the promoters of its target genes (Figure 2C) [141]. Through a similar molecular mechanism, tsRMST suppressed the expression of the WNT family member 5A (WNT5A), thus impairing epithelial-to-mesenchymal transition (EMT) and in vitro differentiation of hESC (Figure 2C) [142].

Importantly, a few years before the tsRMST chimera was identified, high expression levels of a long non-coding RNA (lncRNA), deriving from the same RMST gene locus, and hereafter referred as lncRMST, was observed in dopaminergic neuronal precursors of murine midbrain [143] and to be upregulated during neuronal differentiation [144]. Interestingly, the lncRMST displays an opposite function with respect to tsRMST. In particular, lncRMST was shown to positively regulate neurogenesis by acting as co-transcriptional regulator of SRY-box transcription factor 2 (SOX2) [144]. Although the transcription factor SOX2 is known to regulate a transcriptional program essential for the maintenance of stem cell self-renewal, and in turn, neural stem cell fate [145], when it is bound by lncRMST it promotes the expression of neurogenic transcription factors (i.e., achaete-scute complex-like 1 (ASCL1), neurogenin 2 (NEUROG2), hairy/enhancer-of-split related with YRPW motif 2 (HEY2) and distal-less homeobox 1 (DLX1), and contributes to neuronal differentiation [144].

The interplay between ts-RMST and lincRMST represents a clear example of how canonical and unconventional splicing events from the same gene locus can contribute to the functional diversity that underlies brain complexity by generating multiple transcript variants that regulate overlapping or opposite functions.

### 4.4. Recursive Splicing in Brain Physiology

Recursive splicing is more frequent in human brain than in other tissues, with RS exons being most abundant in long first introns of neuronal-specific genes [6]. Furthermore, numerous transcripts generated by recursive splicing encode for proteins with functional roles in axon guidance and cell adhesion [6]. Thus, recursive splicing ensures high-fidelity splicing of long introns in neuronal transcripts, probably contributing to guaranteeing that the correct program of neuronal development and differentiation occurs. Nevertheless, the functional significance of recursive splicing in the CNS and its impact on brain physiology still remain to be fully elucidated.

Notably, deficient activity of the EJC in mice—a complex that acts as negative regulator of recursive splicing [80,82,83]—has the strongest effect on brain, where it leads to skipping of RS exons in genes with neurodevelopmental functions, likely contributing to a microcephaly phenotype [82]. Indeed, skipped RS exons were also observed in assembly factor for spindle microtubules (Aspm) and centromere protein j (Cenpj) [82], two genes previously associated with autosomal recessive microcephaly that encode proteins essential for mitotic cell progression during embryonic neurogenesis [146].

Interestingly, several other human neurodevelopmental disorders are associated with mutations in EJC components [147]. It is plausible that dysregulation of recursive splicing, as result of RS site mutations and/or loss of function mutations in EJC components, may contribute to neurodegenerative disease and brain cancers. It is noteworthy that multistep splicing might also contribute to the expression of disease-related circular transcripts, as in the case of the sphingomyelin synthase gene [148]. In light of this, it is conceivable that the involvement of recursive splicing in brain physiopathology is still underestimated and only the advent of new technologies and more targeted research (i.e., transcriptomic data analysis of nascent RNAs rather than of RNAs at steady state [149]) will clarify this issue.

Due to the lack of experimental evidence on the role of recursive splicing in brain tumours, the corresponding paragraph on this topic in the next and last section of the review will not be available.

## 5. Impact of Non-Canonical Splicing in Brain Tumours

### 5.1. circRNAs in Brain Tumours

The evolution of transcriptomic and bioinformatic analyses has allowed researchers to efficiently identify non-canonical splicing events in several tumours [4,150,151]. In particular, circRNAs have been implicated in numerous cancers, where they are often aberrantly expressed. A growing number of studies highlights the importance of circRNAs both as active players involved in different aspects of cancer biology and as potential tools for diagnosis and prognosis, thereby exploiting their specific expression in cancer tissues [152].

The abundance of circRNAs in the human brain [8,153] and the specificity of their expression patterns in different brain regions have raised growing interest in the role of their dysregulation in the pathogenesis and progression of brain cancers [154,155,156,157], including gliomas and medulloblastomas, which represent the most frequent adult and paediatric CNS tumours, respectively [158].

#### 5.1.1. CircRNAs in Gliomas

Gliomas originate from glial cells and affect brain and spinal cord. The astrocytic lineage generates the most frequent variant of gliomas, glioblastoma multiforme (GBM), which is associated with very poor prognosis [158]. Based on their capacity to act as competing endogenous RNAs (ceRNAs), circRNAs were shown to inhibit miRNA-driven repression of target genes. For instance, circNT5E (5′-nucleotidase ecto) acts as ceRNA for miR-422a and other miRNAs, thus promoting proliferation and invasion of GBM cells [159]. Similarly, circ0046701 and circHIPK3 fostered proliferative and invasive features of glioma cells by targeting the axes miR-142-3p/ITGB8 (integrin subunit beta 8) and miR-654/IGF2BP3 (insulin-like growth factor 2 mRNA-binding protein 3), respectively [160,161,162].

Upregulation of signalling pathways involved in cellular growth and proliferation is a key event during cancer development [163] and represents a mechanism through which circRNAs affect the onset and progression of tumours. Via sponging miR-422a, circNT5E sustained the activation of the phosphatidylinositol-4,5-bisphosphate 3-kinase/akt (PI3K/AKT) serine/threonine kinase pathway [159], while circ0014359 stimulated the same pathway by regulating miR-153 [164]. Furthermore, it has been demonstrated that circNFIX (nuclear factor I X) was able to promote Notch signalling by sponging miR-34a-5p, a miRNA that targets the NOTCH1 receptor [165].

Several recent studies also link the expression of circRNAs with tumour angiogenesis in gliomas [166,167,168]. He et al. demonstrated that, circSHKBP1 (SH3KBP1-binding protein 1) promoted the expression of the pro-angiogenic factors forkhead box p1/2 (FOXP1/FOXP2) in glioma-exposed endothelial cells by targeting miR-544a/miR-379 [166]. Similarly, in the same in vitro model, circ002136 sustained glioma angiogenesis by regulating SOX13 (SRY-box transcription factor 13) through inhibition of miR-138-5p [168].

In addition to miRNAs, circRNAs can also directly bind RBPs and affect their activity. For instance, circSMARCA5 (SWI/SNF related, matrix associated, actin-dependent regulator of chromatin, subfamily a, member 5) was shown to interact with the splicing factor SRSF1 and to inhibit its activity [169]. SRSF1 is upregulated in GBM patients and promotes splicing of pro-angiogenic vascular endothelial growth factor (VEGF) isoforms. CircSMARCA5 counteracts this activity and plays a tumour suppressor role by inhibiting the SRSF1-dependent cancer-related splicing program. Coherently, circSMARCA5 downregulation was observed in high-grade GBM [169,170]. Recently, Bronisz et al. identified an RNA-protein complex comprising Dicer 1 Ribonuclease III (DICER) RNA-binding motif protein 3 (RBM3) and circ2082 that regulates the maturation of the global miRNAome in GBM cells (Figure 3A). Upregulation of circ2082 was associated with the aberrant nuclear localization of DICER—the major enzymatic complex responsible for miRNA maturation—resulting in dysfunctional and pro-tumorigenic regulation of the microRNA processing machinery. Of note, circ2082 depletion restored DICER cytoplasmic localization and a pre-malignant miRNA expression signature, with concomitant mitigation of the tumorigenic phenotype in vitro and in vivo [171]. This evidence supports a pro-tumoral role for circ2082 in conferring molecular identity and tumorigenic potential to GBM cells through a pervasive DICER-dependent dysregulation of miRNA expression profiles (Figure 3A).

Some circRNAs can be also translated through cap-independent mechanisms to produce small peptides [172]. For instance, circSHPRH (SNF2 histone linker PHD RING helicase) encodes a 17 kDa protein (SHPRH-146aa) that acts as a decoy to protect its SHPRH linear counterpart from ubiquitination and subsequent degradation [72]. Interestingly, both circSHPRH RNA and encoded protein were strongly downregulated in GBM cells. The lower expression levels of SHPRH-146aa in GBM cells resulted in reduced expression of the linear SHPRH transcript and protein, which, in turn, is necessary for the ubiquitination of the pro-proliferative factor proliferating cell nuclear antigen (PCNA). Thus, lower expression of the linear SHPRH transcript resulted in a higher concentration of PCNA and higher proliferative rate of GBM tumour cells [72]. On the other hand, circFBXW7 (F-Box and WD repeat domain containing 7) encodes for a protein of 185 residues (FBXW7-185aa) that binds and inhibits the de-ubiquitinating enzyme ubiquitin-specific peptidase 28 (USP28), thus promoting c-MYC ubiquitination/degradation and cell cycle arrest [173]. In line with this tumour suppressor function, both circFBXW7 and FBXW7-185aa are less expressed in GBM, whereas high levels of circFBXW7 are positively correlated with patient survival [173].

Tumour suppressor activity through the encoded protein has also been shown by two other circRNAs. CircPINT (long intergenic non-protein-coding rna, p53-induced transcript) exon 2-derived PINT-87aa was found to inhibit the transcriptional elongation of oncogenes’ transcripts, such as SOX2 and c-MYC, by interacting with the polymerase-associated factor 1 complex (PAF1c) [174]. More recently, Xia et al. showed a role for the circAKT3-encoded AKT3-174aa protein in reducing GBM progression. CircAKT3 is generated from the human AKT serine/threonine kinase 3 (AKT3) locus and encodes for a peptide of 174 amino acids, which acts as a dominant negative variant of AKT. AKT3-174aa was shown to compete with AKT for the binding to phosphorylated phosphoinositide-dependent kinase-1 (PDK1), an essential kinase which directly phosphorylates AKT at threonine residue 308 (Thr308) mediating AKT activation [175]. Activation of the AKT signalling pathway contributes to cancer cell survival and proliferation and is considered a hallmark for several tumours. Thus, AKT3-174aa exerted a tumour-suppressive role in GBM by acting as negative regulator of AKT activity, leading to significant reduction in cell proliferation, clonogenic activity and radiation resistance [175]. Interestingly, AKT3-174aa expression is negatively correlated with the tumorigenicity of GBM cells in pre-clinical studies and positively correlated with overall survival in GBM patients [175].

An interesting oncogenic circuitry involving a circRNA has been recently uncovered. The circRNA, termed circE-Cad, is generated from the E-cadherin locus (CDH1) and is upregulated in glioma stem cells and glioma tissues compared to controls. CircE-Cad encodes for a peptide of 255 amino acid residues (C-E-Cad), which is secreted by glioma cells and stimulates the epidermal growth factor receptor (EGFR) [176]. In turn, EGFR triggers the activation of signalling cascades impinging on signal transducer and activator of transcription 3 (STAT3), AKT, and the mitogen-activated protein kinases (MAPKS) ERK1/2, thereby promoting the oncogenic phenotype of glioma cells. In support of its functional relevance, circE-Cad expression is associated with reduced survival in GBM patients [176].

In conclusion, circRNAs are emerging as valuable molecules that could serve as both prognostic biomarkers and molecular targets for GBM and glioma patients. Moreover, the circRNAs-derived peptides might represent cancer-specific neoepitopes whose immunogenic potential could be exploited for immunotherapeutic approaches in the near future. This, and other aspects of the translational impact of circRNAs and circRNA-derived peptides, will be discussed later in the review.

#### 5.1.2. CircRNAs in Medulloblastoma

Medulloblastoma (MB) is the most common malignant brain tumour in children [158]. MB generally affects the cerebellum, and based on the molecular signature, age of onset and prognosis, is classified into four subgroups: WNT, sonic hedgehog (SHH), Group 3 (G3) and Group 4 (G4). Among them, SHH is the most common molecular subgroup in adults, whereas Group 3 is the most aggressive subtype in infancy and early childhood [158]. Several studies have demonstrated that splicing defects often underlie MB pathogenesis [177,178,179]. An A > G mutation at the third nucleotide of U1 snRNA has been frequently detected in SHH-MB subgroup patients. The mutation impairs proper recognition of the 5′ splice sites and globally affects the splicing signature of MB cells. Since the U1 snRNA mutation is correlated with poor prognosis, it is conceivable that splicing dysregulation acts as a potent oncogenic driver in this cohort of MB tumours [177].

A few studies have also investigated the role of non-canonical splicing in MB [180,181,182], mostly focusing on the biological relevance and the diagnostic value of circRNAs. For instance, a signature of 33 differentially expressed circRNAs was identified in MB samples compared to normal cerebellum. The same study also showed the involvement of circSKA3 (spindle and kinetochore associated complex subunit 3) and circDTL (denticleless E3 ubiquitin protein ligase homolog) in the proliferation of MB cells [180], providing a functional link with the disease. By contrast, downregulation of a sizeable fraction (~25%) of the 29 differentially expressed circRNAs in SHH-MB tumours and MB cell lines (e.g., DAOY) did not exert a significant impact on cell proliferation in vitro or on expression of their linear transcript counterpart [181]. Although no other biological function was investigated, this experimental evidence indicates that, even if they are strongly deregulated in a tumour context, circRNAs do not necessarily play a key role in tumorigenesis. CircRNAs might represent, in some cases, simple “passenger molecules” with limited functional impact, while their production might reflect the stochasticity of the general deregulation of transcriptional and/or post-transcriptional processes in tumour cells. In support of this hypothesis, linear back-splicing events (exon repetition), derived from intermolecular splicing reactions between two or more immature transcripts of the same gene, have been identified in the same SHH-MB context [181]. Thus, despite relative abundance and differential expression, considerable caution must be taken when addressing the functional roles of circRNAs in tumour cell growth.

Although their biological function is not always well-established, the expression profiles of multiple circRNAs (circRNAome) might contribute to defining specific molecular tumour subgroups. For instance, a very recent article highlights the discriminating power of circRNA signatures in distinguishing MB subgroups [182]. By using the 500 most differentially expressed circRNAs in two different cohort of MB patients, authors were able to distribute patients into the four core MB subgroups (WNT, SHH, G3, and G4), thus confirming the same classification obtained by combining proteomic, transcriptomic, and methylomic data [182] Furthermore, they also identified the abnormal overexpression of circRMST (rhabdomyosarcoma 2-associated transcript) as a specific and highly reproducible biomarker in WNT medulloblastomas [182].

Thus, a fascinating scenario is emerging in which the complexity of circRNAs’, and more generally, ncRNAs’, networks in human brain physiology and pathology can provide new opportunities for a better understanding of diseases and for future translational applications.

### 5.2. Splicing of Microexons in Brain Tumors

The expression of SRRM4—the master regulator of neuronal microexons (see above)—is downregulated in several types of cancers [183]. Importantly, the lower expression of SRRM4 is correlated with repression of SRRM4-dependent microexon splicing and with enhanced proliferative and self-renewing abilities of cancerous cells. These effects may result from removal of the neuronal differentiation normally elicited by SRRM4 (Figure 3B) [183]. Even if the functional role of individual microexons in proliferation and tumorigenesis was not investigated, their concomitant deregulation likely affected the activity of multiple genes with known roles in brain tumours [183]. For instance, among the SRRM4 target genes, dedicator of cytokinesis 7 (DOCK7) and prosaposin (PSAP) are known to be involved in cell invasion and tumorigenesis, respectively, in GBM [184,185]. In support of a possible anti-proliferative role for these microexons, their inclusion is negatively correlated with the mitotic index across multiple tumour types (Figure 3B) [183].

Although the functional role of SRRM4 has been well-described and well-characterized in the trans-differentiation and progression of neuroendocrine prostate cancer [186,187], to date, limited experimental evidence attests a key role for SRRM4 in brain tumorigenesis. However, recent findings indicated that SRMM4 is the only SRRM family member whose expression levels were significantly decreased in GBM compared to normal brain tissue, and its low expression was significantly correlated with poor survival of GBM and low-grade glioma patients [188]. These observations suggest that the expression of SRRM4 and the execution of a specific SRRM4-dependent microexons splicing program might represent a brake that prevents the onset and/or progression of these brain tumours.

Coherently, also the expression of two other RBPs, RBFOX2 and PTBP1, known regulators of microexon splicing [76], showed altered expression in colorectal cancer (CRC) [189]. Interestingly, both RBPs contributed to the metastatic phenotype of CRC through the regulation of microexon splicing [189].

Despite the increasing evidence for microexons impacting cellular physiology within the CNS, mechanistic details illustrating their functional importance in brain tumours is still limited. Due to their small size, microexons necessitate specialized computational approaches to be detected. The advent of novel technologies and bioinformatics pipelines in the last few years has helped to identify numerous microexons that were previously uncharacterized. For instance, MicroExonator is a novel pipeline based on the detection of sequences inserted between annotated splice sites, which allows for the identification of microexons that were not previously reliably detected by typical RNA sequencing aligners [78]. Interestingly, analyses employing this bioinformatics tool produced the most comprehensive catalogue of microexons available to date, highlighting distinct inclusion patterns during mouse neuronal development and cell type-specific expression profiles.

The current knowledge on the role played by splicing of microexons in brain tumorigenesis likely represents the tip of the iceberg. Future work aimed at dissecting the functional impact of these extremely small genetic units on protein functions, and therefore on specific biological processes, is required in order to fully understand their implication in oncogenesis.

### 5.3. Trans-Splicing in Brain Tumors

Although there is limited experimental evidence that correlates chimeric RNA formation with the onset and/or progression of brain tumours, their contribution in numerous other tumour contexts is greatly emerging [190]. However, a very recent paper by Shi and colleagues [191] clearly showed how brain tumour cells may exploit trans-splicing-mediated fusion RNAs to their own advantage in order to guarantee their proliferation and survival [191]. By analysing a cohort of 172 RNA-sequenced neuroblastoma tumours, the authors identified more than 900 primarily intra-chromosomal fusion RNAs, with most of them specifically expressed in neuroblastoma [191]. Interestingly, the identified fusion RNAs included transcripts from well-known oncogenes, previously shown to be important for neuroblastoma biology (i.e., MYCN and LIM domain only 1 LMO1), and they involved chromosomal regions commonly gained or lost in high-risk neuroblastoma [191]. The identified RNA chimeras contained canonical splicing donor and acceptor sites and the fusion junctions were located in chromosomic regions no larger than 100 Kb, supporting the primary role of trans-splicing events for their biogenesis [191]. Importantly, Pladienolide B-dependent splicing inhibition strongly reduced the expression of fusion transcripts, whereas the expression of the linear and independent transcripts was largely unaffected [191]. These observations open exciting scenarios as if to suggest that in a tumour context, such as that of neuroblastoma, the splicing decisions versus conventional or unconventional events are finely regulated by different and currently unknown rules and regulatory mechanisms directly orchestrated by tumour cells to their own advantage. Coherently, the chimeric RNA zinc finger protein 451 (ZNF451)-BAG cochaperone 2 (BAG2) fusion, highly expressed in neuroblastoma, generated a novel oncogenic protein with distinct protein–protein binding properties compared to wild-type BAG2 (Figure 3C) [191]. Indeed, ZNF451-BAG2 chimera spans the 3′ UTR of ZNF451 and the second exon of BAG2, potentially generating a truncated BAG2 transcript lacking the coiled-coil domain located in the first exon. Interestingly, ZNF451-BAG2 fusion transcript encoded a smaller 19.6 kDa protein (ΔBAG2) that impaired the clearance of the phosphorylated forms of the TAU protein (pTAU), as consequence of the impaired binding to heat shock cognate 70 (HSC70), and inhibited retinoic acid-induced differentiation of neuroblastoma (Figure 3C) [191]. Altogether, these observations suggest probable implications for ZNF451-BAG2, and, more generally, for trans-splicing in age-related neurodegenerative diseases, such as tauopathies, and as has already been demonstrated for other RNA fusions, such as tsRMST chimera [141], in the maintenance of stemness potential.

Recently, a novel trans-splicing-mediated fusion transcript generated from the enhancer of polycomb homolog 2 (EPC2) and GULP PTB domain containing engulfment adaptor 1 (GULP1) loci has been identified in MB [192]. Although the EPC2-GULP1 chimera did not show MB subgroup-specificity, it was detected in 3 out of 11 MB patients analysed and not in other paediatric brain tumours [192]. Importantly, the peptide encoded by the EPC2-GULP1 fusion transcript presented immunogenic potential, as demonstrated by its ability to induce a CD8+ T-cell-mediated immune response [192]. This observation has extremely relevant clinical implications, suggesting that neo-antigens depending on trans-splicing for their biogenesis, as is the case for EPC2-GULP1 fusion peptide, may represent optimal candidates for immunotherapy, especially in a tumour context, such as that of MB, with minimal mutational load and low immunogenicity [193].

In conclusion, although numerous chimeric transcripts with interesting implications in the biology of MB have been identified [193,194,195,196], it is conceivable that a substantial part of them is derived from trans-splicing events or that, if expressed in normal tissue, may have contributed to the formation of the gene fusions.

## 6. Therapeutic Applications of Non-Canonical Splicing

Targeting splicing mechanisms represents an emerging therapeutic possibility for several human diseases, including cancer [157,197,198]. Common approaches include RNA editing through single-stranded DNA molecules and the use of small-molecule compounds acting on splicing factors’ functions [198]. Antisense oligonucleotides (ASOs), called splice-switching oligonucleotides (SSOs), promote exon skipping or inclusion by binding to the complementary target pre-mRNAs [198]. In GBM, these strategies have been used to impair the splicing outcomes of the two oncogenic proteins MAPK interacting serine/threonine kinase 2 (MNK2) and human telomerase reverse transcriptase (hTERT), inhibiting the junctions and thus hindering tumour growth [199,200]. On the other hand, continuous efforts have been made in the development of chemical inhibitors targeting spliceosome’s components, some of them with demonstrated oncogenic roles in brain tumours. Efficient splicing in GBM is highly dependent on protein arginine methyltransferase 5 (PRMT5), which catalyses methylation of several spliceosomal proteins [201,202,203]. Interestingly, effective PRMT5 inhibitors have been developed and are currently in clinical trial [204]. Recently, two additional molecules (GSK591 and LLY-283) have been identified as potent inhibitors of PRMT5 in GBM patient-derived cancer stem cell lines, resulting in altered splicing and impaired clonogenic capacity [205].

Events and mechanisms of non-canonical splicing have increasingly become an attractive opportunity for the identification of new biomarkers in cancer and for the development of innovative strategies in cancer therapy.

Due to their stability, relative abundance in body fluids and tissue specificity, circRNAs have been proposed as prognostic and diagnostic biomarkers in different tumours, including gliomas, GBM and MB [173,176,182,206]. Besides these findings, there is growing evidence highlighting the potential of circRNAs as therapeutic tools and targets by modulation of their expression using different strategies, such as ASOs, RNA interference, CRISPR/Cas technology or viral vectors [207]. CRISPR/Cas9 has been successfully used to generate KO mice for CDR1as, a circRNA involved in synaptic transmission [9], while overexpression of the tumour suppressor circITCH (Itchy E3 ubiquitin protein ligase) elicited protective effects against doxorubicin-induced cardiotoxicity, with potential benefits for cancer patients treated with this anti-tumour agent [208].

Inhibiting oncogenic miRNAs is another interesting strategy where the functions of circRNAs may be exploited (Table 1). For example, a synthetic circRNA that sponges miR21 caused the upregulation of the death domain-associated protein (DAXX) and demonstrated anti-proliferative activity in gastric cancer [209]. On the other hand, the coding potential of circRNAs has a fascinating perspective as a tool to deliver therapeutic proteins in vivo. By using a ribozymatic method or a recombinant adeno-associated virus, two different groups have efficiently generated and delivered artificial circRNAs encoding different types of proteins in cells and tissues [210,211]. Interestingly, as different factors such as diet, metabolism and hypoxia affect miRNAs regulation in cancer [212,213], the modulation of the aberrant miRNAs’ expression pattern through particular diet regimens is gaining interest in the scientific community as a potential therapeutic approach for the treatment of brain tumours [213,214].

Mechanisms of trans-splicing have been recently investigated as potential targets to obtain a functional correction of desired RNAs. Conveniently engineered trans-splicing molecules that contain the chosen splicing domains were delivered to cells, and by exploiting the nuclear splicing machinery (SMaRT, spliceosome-mediated RNA trans-splicing) or internal group I intron ribozymes, they were able to trans-splice the corrected coding sequences into target pre-mRNAs [215,216,217]. Both methods have been used to restore the wild-type functions of the tumour-suppressor proteins p16 and p53 in different models of solid tumours [218,219,220,221]. Furthermore, the SMaRT method has also been employed to correct mutations in genetic diseases such as cystic fibrosis (CF) and haemophilia A. In CF, different SMaRT strategies have demonstrated to be efficient in correcting the ΔF508 mutation in CF transmembrane conductance regulator (CFTR) gene [222,223,224,225], while in a haemophilia A knockout mice model, a pre-trans-splicing molecule has been able to rectify endogenous FVIII (coagulation factor VIII) mRNA [226].

The relative plasticity of the abovementioned techniques also allows us to exploit trans-splicing reactions by other approaches. In the suicide gene therapy approach, inactive molecules are converted to active cytotoxic compounds in cancer cells via proteins encoded by exogenous trans-splicing sequences. To obtain specificity toward tumour cells, the trans-splicing reaction is targeted to genes that are overexpressed in cancer [215]. For example, a system using Herpes simplex virus thymidine kinase-ganciclovir has been trans-spliced into the transcript of hTERT [215]. Interestingly, additional selectivity can be introduced by adding tissue-specific promoters or responsive elements regulating the exogenous expression of the trans-splicing RNA molecules [227,228,229,230,231,232]. A second approach is based on the trans-splicing reaction between two pre-mRNA segments delivered to cancer cells and encoding the 5′ and 3′ fragments of toxins. Using this method, cell death has been successfully induced in epidermolysis bullosa-associated squamous cell carcinoma cells, where the streptolysin O coding sequence was introduced at the 3′ of the matrix metalloproteinase-9 (MMP9) pre-mRNA [233].

Targeting tumour-specific antigens by immunotherapy is a new and promising frontier in the effort to develop efficient and personalised anti-cancer therapies. By using a proteogenomic approach that includes whole-genome sequencing, RNA-seq, and mass spectrometry, 10s of neoantigens were identified in MB tumours, most of which originated from aberrant splice junctions. Of note, these peptides were immunogenic and generated functional T-cell populations (CD4 and CD8) that were specifically reactive against MB cells in vitro [234]. These results are consistent with the recently discovered mutations in U1 small nuclear RNA, which generate a dysfunctional splicing machinery in SHH MB tumours [177]. Indeed, the abnormal splicing program and the consequent anomalous splicing junctions potentially represent a source of new peptides with immunogenic properties.

## 7. Concluding Remarks

The increased understanding of the mechanisms underlying the generation of non-canonical spicing junctions and the simultaneous advances in sequencing and bioinformatics techniques have significantly improved our knowledge about the splicing landscape in brain tissues, both in physiological and pathological conditions. Mounting evidence demonstrates that non-canonical splicing in CNS tumours plays a crucial role, affecting different aspects of cancer cell biology. Furthermore, both back-splicing and trans-splicing have encouraging translational applications that span from their potential use as diagnostic and prognostic biomarkers to the development of new therapeutic tools. Although challenges are still in place in terms of off-target effects and delivery issues, these findings strongly suggest the possible setup of personalized therapies in brain tumours that will exploit the molecular properties of such a particular kind of splicing regulation.

## Figures and Tables

**Figure 1 ijms-23-02811-f001:**
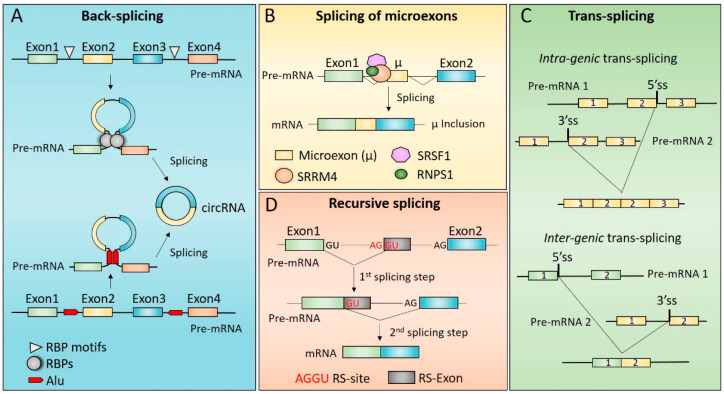
Mechanisms of non-canonical splicing. (**A**) Pre-mRNAs circularization can be promoted by inverted Alu repeats- or RBPs dimerization-mediated base-pairing between upstream and downstream introns flanking circulating exons. (**B**) Inclusion of microexons (µ) is positively regulated by RBPs, such as SRRM4, RNPS1 and SRSF1, which favour spliceosome assembly on splice sites. (**C**) Two individual pre-mRNAs transcribed from the same gene can be spliced leading to an mRNA with an exon duplication (intragenic trans-splicing). Moreover, transcripts from different genes can be spliced to generate a chimeric RNA (intergenic trans-splicing). (**D**) During recursive splicing, long introns are removed in a two-step process mediated by the RS site, containing a 3′ splice site dinucleotide (AG) followed by a 5′ splice site dinucleotide (GU). In the first splicing reaction, the 3′ splice site of the RS site is used to remove the upstream part of the intron. The second splicing reaction uses the 5′ splice site of the RS site to remove the downstream part of the intron. Some recursively spliced introns contain an RS exon that is removed during the second step of the recursive splicing by usage of the new 5′ splice site generated by exon–RS exon junction.

**Figure 2 ijms-23-02811-f002:**
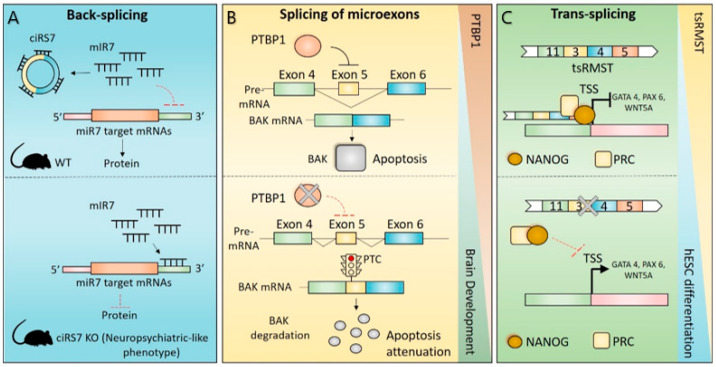
Functional role of non-canonical splicing in brain physiology. (**A**) CDRas1/ciRS-7 acted as a sponge for miR7 allowing for the expression of miR7-target genes. In CDRas1/ciRS-7 KO mice, miR7-target genes are downregulated resulting in defects in sensorimotor gating. (**B**) During brain development, downregulation of the splicing factor PTBP1 allowed for the neural-specific inclusion of microexon 5 in BAK1 mRNA. The inclusion of this microxon triggered NMD of Bak1 transcripts, leading to reduced expression of pro-apoptotic BAK1 protein and neuron survival. (**C**) The trans-spliced chimera generated by the RMST locus (tsRMST) guaranteed pluripotency of hESC by suppressing the expression of differentiation-related genes, such as GATA4, PAX6 and WNT5A, through the recruitment on their promoter of the transcription factor NANOG and PRC.

**Figure 3 ijms-23-02811-f003:**
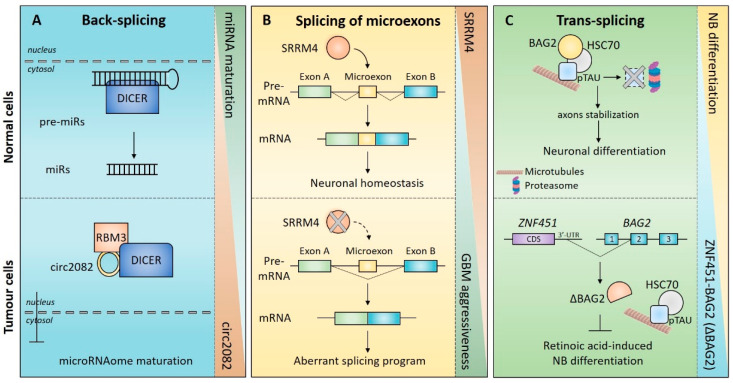
Functional role of non-canonical splicing in brain tumours. (**A**) Upregulation of circ2082 in GBM cells impairs the regular miRNA processing by sequestering DICER in the nucleus of cancer cells. The resulting miRNA maturation machinery generates an aberrant miRNAome that drives tumorigenesis. (**B**) Reduced expression of the main regulator of neuronal microexons SRRM4 is associated with aggressive GBM. Several pieces of evidence across different tumours links the abnormal microexons’ splicing with enhanced proliferation and mitotic index. (**C**) In neuroblastoma cells (NB), the differentiation program, involving the complex BAG2/HSC70 on microtubules, is impaired by a trans-splicing event between the 3′ UTR of ZNF451 mRNA and the second exon of BAG2 mRNA, which generates a fusion transcript encoding a truncated BAG2 protein (ΔBAG2). ΔBAG2 is unable to bind HSC70 and subsequently unable to promote the degradation of the phosphorylated form of TAU.

**Table 1 ijms-23-02811-t001:** Examples of miRNAs whose activity is regulated by circRNAs in brain.

circRNAs	miRNAs	Dysregulation	Downstream Genes and Signaling Pathway Affected	Phenotype	Refs.
ciRS-7	miR-7	down	UBE2A	Neuropsychiatric-like phenotype	[53,54,235]
circNT5E	miR-422a	up	PI3K/AKT signaling	Proliferation, Invasion	[159]
circ0046701	miR-142-3p	up	ITGB8	Proliferation, Invasion	[160]
circHIPK3	miR-654	up	IGF2BP3	Proliferation, Invasion	[55,161]
circ0014359	miR-153	up	PI3K/AKT signaling	Proliferation, migration, Invasion, apoptosis	[164]
circNFIX	miR-34a-5p	up	Notch signaling	Proliferation, migration, Invasion, apoptosis	[165]
circSHKBP1	miR-544a miR-379	up	FOXP1/FOXP2/AGG1 PI3K/AKT and ERK signaling	Proliferation, migration, angiogenesis	[166]
circ002136	miR-138-5p	up	SOX13/SPON2	Migration, invasion angiogenesis	[168]

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
