# Peer review of "Non-Canonical Splicing and Its Implications in Brain Physiology and Cancer"

_ijms, 2022, doi:10.3390/ijms23052811_

Round 1

Reviewer 1 Report

The authors propose a very interesting literature review of a very new and interesting topic.

The work is well presented with a good basis of literature and a good examination of the mechanisms of action.

In my opinion, to complete it, it would be necessary to:

- Analyze which miRNAs (in addition to mir7) are influenced by circRNAs, perhaps in a summary table, in order to have a possible tool to use as a biomarker

- Based on miRNAs, determine which factors are affected by other factors, other than cancer, such as diet, supplement use, or physical activity; in particular, given the localization in the CNS, it has been seen that the ketogenic diet influences the profile of miRNAs (for example 10.2174 / 2211536608666181126093903)

Author Response

Rebuttal Letter

Reviewer #1

1. Analyze which miRNAs (in addition to mir7) are influenced by circRNAs, perhaps in a summary table, in order to have a possible tool to use as a biomarker

As suggested by the reviewer, we added a summary table in which miRs whose activity is strongly affected by specific circRNAs are listed (Table 1 in the manuscript).

2. Based on miRNAs, determine which factors are affected by other factors, other than cancer, such as diet, supplement use, or physical activity; in particular, given the localization in the CNS, it has been seen that the ketogenic diet influences the profile of miRNAs (for example 10.2174 / 2211536608666181126093903).

As suggested by the reviewer, at the line 1083 of the manuscript, we have highlighted the experimental observations that certain extrinsic and intrinsic factors, such as diet, metabolism and hypoxia, could all contribute to the deregulation of miRs expression profiles in cancer. Thus, interferring with some of these factors could acquire therapeutic value in cancer patients. We have added three references on this topic (Ref. 212-214).

212. Chan, B.; Manley, J.; Lee, J.; Singh, S.R. The emerging roles of microRNAs in cancer metabolism. Cancer Letter, 2015, 356, 301–8.

213. Woolf , E.C.; Syed, N.; Scheck, A.C. Tumor Metabolism, the Ketogenic Diet and β-Hydroxybutyrate: Novel Approaches to Adjuvant Brain Tumor Therapy. Frontiers in Molecular Neuroscience, 2016, 9, 122.

214. Cannataro, R.; Perri, R.; Gallelli, L.; Caroleo, M.C.; De Sarro, G.; Cione, E. Ketogenic Diet Acts on Body Remodeling and MicroRNAs Expression Profile. Microrna, 2019, 8, 116-126.

Reviewer 2 Report

A very well-written, comprehensive review on the role of numerous and different mechanisms of RNA maturation. The number of references cited is significant and the reference list covers the relevant literature adequately. I have no comments and consider this article suitable for publication.

Author Response

Not required